# STACK ATTENTION:
# IMPROVING THE ABILITY OF TRANSFORMERS TO MODEL HIERARCHICAL PATTERNS

**Brian DuSell**[*]
Department of Computer Science
ETH Zürich
`brian.dusell@inf.ethz.ch`

**David Chiang**
Department of Computer Science and Engineering
University of Notre Dame
`dchiang@nd.edu`

## ABSTRACT

Attention, specifically scaled dot-product attention, has proven effective for natural language, but it does not have a mechanism for handling hierarchical patterns of arbitrary nesting depth, which limits its ability to recognize certain syntactic structures. To address this shortcoming, we propose stack attention: an attention operator that incorporates stacks, inspired by their theoretical connections to context-free languages (CFLs). We show that stack attention is analogous to standard attention, but with a latent model of syntax that requires no syntactic supervision. We propose two variants: one related to deterministic pushdown automata (PDAs) and one based on nondeterministic PDAs, which allows transformers to recognize arbitrary CFLs. We show that transformers with stack attention are very effective at learning CFLs that standard transformers struggle on, achieving strong results on a CFL with theoretically maximal parsing difficulty. We also show that stack attention is more effective at natural language modeling under a constrained parameter budget, and we include results on machine translation.

## 1 INTRODUCTION

Although transformers (Vaswani et al., 2017) have proven very successful on natural language, linguists have long held that language contains hierarchical syntactic structures (Chomsky, 1956, *inter alia*). Transformers do not appear to have any explicit mechanism for dealing with nested syntactic patterns of arbitrary depth. Recent theoretical work, in fact, has shown that finite-precision transformers cannot recognize certain syntactic patterns, such as Dyck-2 (Hahn, 2020), although they can do so up to bounded depth (Yao et al., 2021). Recent work has also shown that transformers have a linear, rather than hierarchical, inductive bias (Petty & Frank, 2021; Mueller et al., 2022) unless trained long past convergence (Murty et al., 2023a). This might explain, at least partly, why transformers are much less data-efficient than human children at learning language (Gilkerson et al., 2017; van Schijndel et al., 2019; Zhang et al., 2021; Frank, 2023; Warstadt et al., 2023).

In this paper, we propose a novel type of attention mechanism, **stack attention**, that is explicitly designed to model hierarchical patterns. It does this by treating input vectors as items in a stack, and performing a soft-selection not directly over input vectors, but over sequences of stack actions. We motivate this theoretically by pointing out that pushdown automata (PDAs), which are finite automata augmented with stacks, recognize the entire class of context-free languages (CFLs), which capture the essence of compositionality and recursion in natural language syntax (Hopcroft & Ullman, 1979; Sipser, 2013). Accordingly, we expect stack attention to increase transformers' expressive power, in addition to possibly learning language from fewer examples and generalizing to held-out combinations of syntactic patterns in more human-like fashion.

Stack attention draws from prior work on **differentiable stacks**, which are continuous functions that approximate the behavior of discrete stacks (Grefenstette et al., 2015; Joulin & Mikolov, 2015; DuSell & Chiang, 2020). We adapt two differentiable stacks into attention operators: the "superposition" stack of Joulin & Mikolov (2015), and a nondeterministic generalization introduced by

---

[*]Work done while at the University of Notre Dame.

DuSell & Chiang (2023). Previous work attached differentiable stacks to recurrent neural networks (RNNs) as external memory modules, but here we integrate differentiable stacks into transformers in a much deeper way. Whereas standard attention performs a soft-selection over input vectors, these differentiable stacks perform a soft-selection over *sequences of push and pop actions* on stacks of vectors, returning the topmost vector of the stack of the selected sequence. Accordingly, we use differentiable stacks as a drop-in replacement for standard attention, resulting in a transformer with a latent model of syntax that, unlike most prior work, requires no syntactic supervision, and, in the case of nondeterministic stack attention, is able to recognize the entire class of CFLs.

Our experiments show that transformer language models with nondeterministic stack attention learn CFLs very effectively, consistently outperforming baseline transformers, and outperforming even the previously-proposed VRNS-RNN (DuSell & Chiang, 2022) on a CFL with maximal parsing difficulty (Greibach, 1973). We also show that under a constrained parameter budget, transformer language models with stack attention outperform baseline transformers on the Penn Treebank by 4.3 perplexity points, and we include results on a small machine translation task based on the German-English dataset from Europarl v7. Our code is publicly available.[1]

## 2 RELATED WORK

Our method enjoys two key advantages over prior work on syntax-oriented transformers: (1) it does not require syntactic supervision (although it can be extended to leverage it); and (2) it is unidirectional in time, so it can be used in causal language models and decoders. Most prior work requires syntactic supervision (Shiv & Quirk, 2019; Deguchi et al., 2019; Zhang et al., 2020; McDonald & Chiang, 2021; Qian et al., 2021; Sartran et al., 2022; Murty et al., 2023b). Wang et al. (2019b) proposed an unsupervised but bidirectional model. Kim et al. (2017) proposed a framework for structured attention mechanisms that sum over an exponential number of latent structures, similar to our approach. Their work included an attention operator based on the inside-outside algorithm for projective dependency trees, whereas we propose one based on a dynamic programming algorithm for PDAs. A key difference is that, at each timestep, their attention mechanism marginalizes over all latent structures backward *and forward* in time, so it is bidirectional. Our stack attention marginalizes backward in time only.

## 3 BACKGROUND

We start by reviewing attention and differentiable stacks. For any tensor (matrix, vector) $\mathbf{A}$, let $\mathbf{A}[i, j, \ldots]$ indicate the element or sub-tensor indexed by $i, j, \ldots$. Let $\sigma$ be the logistic function.

### 3.1 SCALED DOT-PRODUCT ATTENTION

Attention has become a cornerstone of modern neural network architectures (Cho et al., 2015; Bahdanau et al., 2015; Vaswani et al., 2017). Loosely defined, attention refers to an operation that, given a sequence of **values** $\boldsymbol{v}_1, \ldots, \boldsymbol{v}_n \in \mathbb{R}^{d_v}$, produces an output $\boldsymbol{v}'$ that represents a linear interpolation, or "soft-selection," over $\boldsymbol{v}_1, \ldots, \boldsymbol{v}_n$.

The transformer uses an attention operator called **scaled dot-product attention (SDPA)**. The self-attention sublayer function in a transformer receives input vectors $\boldsymbol{x}'_1, \ldots, \boldsymbol{x}'_n \in \mathbb{R}^{d_{\text{model}}}$ and linearly transforms them to sequences of **query vectors** $\boldsymbol{q}_1, \ldots, \boldsymbol{q}_n \in \mathbb{R}^{d_k}$, **key vectors** $\boldsymbol{k}_1, \ldots, \boldsymbol{k}_n \in \mathbb{R}^{d_k}$, and value vectors $\boldsymbol{v}_1, \ldots, \boldsymbol{v}_n \in \mathbb{R}^{d_v}$. It runs SDPA for each timestep $1 \leq t \leq n$, producing outputs $\boldsymbol{v}'_1, \ldots, \boldsymbol{v}'_n \in \mathbb{R}^{d_v}$. Each $\boldsymbol{v}'_t$ is linearly transformed to an output $\boldsymbol{y}'_t \in \mathbb{R}^{d_{\text{model}}}$.

$$\boldsymbol{q}_t = \boldsymbol{W}_{\text{q}} \boldsymbol{x}'_t \qquad \boldsymbol{k}_t = \boldsymbol{W}_{\text{k}} \boldsymbol{x}'_t \qquad \boldsymbol{v}_t = \boldsymbol{W}_{\text{v}} \boldsymbol{x}'_t \qquad (1)$$

$$\boldsymbol{z}'_t[i] = \frac{\boldsymbol{q}_t \cdot \boldsymbol{k}_i}{\sqrt{d_k}} \quad (1 \leq i \leq n) \qquad \boldsymbol{z}_t = \text{softmax}(\boldsymbol{z}'_t) \qquad (2)$$

$$\boldsymbol{v}'_t = \sum_{i=1}^{n} \boldsymbol{z}_t[i] \, \boldsymbol{v}_i \qquad \boldsymbol{y}'_t = \boldsymbol{W}_{\text{y}} \boldsymbol{v}'_t \qquad (3)$$

---

[1] https://github.com/bdusell/stack-attention

If we prevent SDPA from attending to future timesteps by ending the iterations over $i$ at $t$ instead of $n$, which is necessary for causal language modeling and decoding, we say it is **causally masked**. Each sublayer typically has multiple attention **heads**, applying multiple SDPA operators to the input sequence and adding the results together.[2]

## 3.2 DIFFERENTIABLE STACKS

A **stack** is a container of elements that allows one to **push** and **pop** elements in last-in-first-out order. A **differentiable stack** is a continuous function that approximates the behavior of a stack; in this paper, we assume elements are vectors in $\mathbb{R}^m$. Suppose we start with an empty stack $\mathcal{S}_0$ and perform $n$ operations on it (e.g. pushing and popping), resulting in a sequence of stacks $\mathcal{S}_1, \ldots, \mathcal{S}_n$. In a differentiable stack, we represent the actions as **stack action** vectors $\boldsymbol{a}_1, \ldots, \boldsymbol{a}_n$, where each $\boldsymbol{a}_t$ is a set of action *weights*. The exact set of available actions varies with the particular style of differentiable stack. The inputs also include a sequence of **pushed vectors** $\boldsymbol{v}_1, \ldots, \boldsymbol{v}_n \in \mathbb{R}^m$ that serve as the new elements inserted by push actions. The output of the differentiable stack is a sequence of **stack reading** vectors $\boldsymbol{r}_1, \ldots, \boldsymbol{r}_n$, where each $\boldsymbol{r}_t$ represents the top vector element of the stack after processing the stack actions and pushed vectors up to timestep $t$. Each differentiable stack can be abstracted into a function STACK that incrementally updates the stack, and a function READING that queries the stack reading from it.

$$\mathcal{S}_t = \text{STACK}(\mathcal{S}_{t-1}, \boldsymbol{a}_t, \boldsymbol{v}_t) \qquad\qquad \boldsymbol{r}_t = \text{READING}(\mathcal{S}_t) \qquad (4)$$

The stack readings are differentiable with respect to the stack actions and pushed vectors, so if the stack is incorporated into a neural network, the whole network can still be trained end-to-end with backpropagation, without any need for supervision over stack actions.

### 3.2.1 SUPERPOSITION STACK

The differentiable stack of Joulin & Mikolov (2015) uses a strategy called **superposition**. It simulates fractionally-weighted stack actions by computing three new, separate stacks: one with all elements shifted down (push), kept the same (no-op), and shifted up (pop). The new stack is an element-wise interpolation ("superposition") of these three stacks. Another way of viewing this is that each element is an interpolation of the elements above, at, and below it at the previous timestep, weighted by push, no-op, and pop probabilities respectively. Here, $\boldsymbol{a}_t = [a_t^{\text{push}}\ a_t^{\text{noop}}\ a_t^{\text{pop}}]^T$ is a probability distribution over the three actions, and the stack state $\mathcal{S}_t$ is a matrix $\boldsymbol{V}_t$ containing the stack elements. We implement $\boldsymbol{V}_t = \text{STACK}(\boldsymbol{V}_{t-1}, \boldsymbol{a}_t, \boldsymbol{v}_t)$ as

$$\boldsymbol{V}_t[i] = a_t^{\text{push}}\ \text{ABOVE}_t(i) + a_t^{\text{noop}}\ \text{AT}_t(i) + a_t^{\text{pop}}\ \text{BELOW}_t(i) \quad (1 \leq i \leq t) \qquad (5)$$

$$\text{ABOVE}_t(i) = \begin{cases} \boldsymbol{v}_t & i = 1 \\ \boldsymbol{V}_{t-1}[i-1] & i > 1 \end{cases} \qquad (6)$$

$$\text{AT}_t(i) = \begin{cases} \boldsymbol{V}_{t-1}[i] & i < t \\ \boldsymbol{0} & i = t \end{cases} \qquad (7)$$

$$\text{BELOW}_t(i) = \begin{cases} \boldsymbol{V}_{t-1}[i+1] & i < t-1 \\ \boldsymbol{0} & i \geq t-1. \end{cases} \qquad (8)$$

The stack reading $\boldsymbol{r}_t = \text{READING}(\boldsymbol{V}_t)$ is simply the top vector element $\boldsymbol{V}_t[1]$, and we set $\boldsymbol{V}_0 = [\boldsymbol{0}]$. This stack has quadratic time and space complexity with respect to input length.

### 3.2.2 NONDETERMINISTIC STACK

DuSell & Chiang (2023) recently proposed a differentiable stack based on **pushdown automata (PDAs)**. PDAs are a type of nondeterministic automaton that consists of a finite state machine connected to an infinite stack memory (for a detailed definition, see Appendix A). They recognize exactly the class of CFLs. Every PDA has a finite alphabet of input symbols $\Sigma$, a finite set of states $Q$, and a finite alphabet of stack symbols $\Gamma$. The initial stack consists of $\perp \in \Gamma$, which is a designated bottom symbol.

---

[2]This is presented a little differently from Vaswani et al. (2017), but it is mathematically equivalent.

A PDA has transitions of the form $q, u \xrightarrow{a} r, v$, which signifies that if the PDA is in state $q \in Q$ and has string $u \in \Gamma^*$ on top of the stack, and string $a \in \Sigma \cup \{\varepsilon\}$ can be scanned from the input tape, then it goes to state $r \in Q$, pops $u$, and pushes string $v \in \Gamma^*$ onto the stack. PDAs are *nondeterministic* in the sense that multiple transitions may apply ambiguously to the same machine configuration, in which case all sequences of transitions are explored. Nondeterminism is essential for recognizing all CFLs, because deterministic PDAs are strictly less powerful. We call a sequence of connecting transitions that starts from the initial machine configuration a **run**. We write $\pi \rightsquigarrow t, q, x$ to indicate that run $\pi$ scans exactly $t$ input symbols, ends in state $q$, and ends with symbol $x$ on top of the stack.

The differentiable stack of DuSell & Chiang (2023) assumes the PDA is in the following normal form, which is equivalent in recognition power. Transitions must have one of the following forms, where $q, r \in Q$, $a \in \Sigma$, and $x, y \in \Gamma$:

$$q, x \xrightarrow{a} r, xy \qquad \textbf{Push } y \text{ on top of } x. \qquad (9)$$

$$q, x \xrightarrow{a} r, y \qquad \textbf{Replace } x \text{ with } y. \qquad (10)$$

$$q, x \xrightarrow{a} r, \varepsilon \qquad \textbf{Pop } x. \qquad (11)$$

**Weighted PDAs (WPDAs)** are an extension of PDAs that assigns a non-negative weight to each transition. The weight of run $\pi$, which we denote $\psi(\pi)$, is the product of its transition weights. DuSell & Chiang (2023) extended WPDAs further by augmenting every instance of a stack symbol with a vector in $\mathbb{R}^m$, so that stack elements are members of $\Gamma \times \mathbb{R}^m$. The purpose of adding vectors is to allow the automaton to transmit information via the stack efficiently using embedding vectors rather than a finite alphabet of discrete symbols. We will refer to it as the **vector PDA (VPDA)**.

Let $w = w_1 \cdots w_n \in \Sigma^n$ be a string of input symbols, and assume that there is a sequence of pushed vectors $\boldsymbol{v}_0, \ldots, \boldsymbol{v}_n$. In a vector PDA, the initial stack consists of $(\bot, \boldsymbol{v}_0)$. Transitions have the following semantics when scanning $w_t$:

$$q, x \xrightarrow{w_t} r, xy \qquad \text{If } (x, \boldsymbol{u}) \text{ is on top, push } (y, \boldsymbol{v}_t).$$

$$q, x \xrightarrow{w_t} r, y \qquad \text{If } (x, \boldsymbol{u}) \text{ is on top, replace it with } (y, \boldsymbol{u}).$$

$$q, x \xrightarrow{w_t} r, \varepsilon \qquad \text{If } (x, \boldsymbol{u}) \text{ is on top, pop it.}$$

Note that transitions can only be conditioned on $(q, x) \in Q \times \Gamma$, never the value of vector $\boldsymbol{u}$, so only $Q$ and $\Gamma$ direct the nondeterministic branching of the VPDA (this will allow Eq. (12) to be tractable).

The differentiable stack of DuSell & Chiang (2023) is a differentiable simulation of a vector PDA. We will refer to it as the **differentiable vector PDA (dVPDA)**. Here, each $\boldsymbol{a}_t$ is the flattening of a tensor $\boldsymbol{\Delta}_t$ which contains the transition weights used when scanning $w_t$. We denote the weight of transition $q, x \xrightarrow{w_t} r, v$ as $\boldsymbol{\Delta}_t[q, x \rightarrow r, v]$. Since $q, r \in Q$, $x \in \Gamma$, and there are $2|\Gamma| + 1$ possibilities for $v$ given Eqs. (9) to (11), the size of $\boldsymbol{\Delta}_t$ is $|Q| \times |\Gamma| \times |Q| \times (2|\Gamma| + 1)$. We set $\boldsymbol{v}_0 = \sigma(\boldsymbol{w}_{\mathrm{v}})$, where $\boldsymbol{w}_{\mathrm{v}} \in \mathbb{R}^m$ is a learned parameter.

Let $\boldsymbol{v}(\pi)$ denote the top stack vector at the end of VPDA run $\pi$. The stack reading $\boldsymbol{r}_t \in \mathbb{R}^{|Q| \cdot |\Gamma| \cdot m}$ includes, for each $(r, y) \in Q \times \Gamma$, an interpolation of $\boldsymbol{v}(\pi)$ for every run $\pi \rightsquigarrow t, r, y$, normalized by the weight of all runs. Let $\boldsymbol{r}_t[r, y]$ denote the slice of $\boldsymbol{r}_t$ corresponding to $(r, y)$. Then

$$\boldsymbol{r}_t[r, y] = \frac{\sum_{\pi \rightsquigarrow t, r, y} \psi(\pi) \, \boldsymbol{v}(\pi)}{\sum_{r' \in Q} \sum_{y' \in \Gamma} \sum_{\pi \rightsquigarrow t, r', y'} \psi(\pi)}. \qquad (12)$$

In this way, the dVPDA is a soft-selection over VPDA runs that outputs the expected top stack vector. Although Eq. (12) sums over an exponential number of runs, it can be computed in cubic time and quadratic space using Lang's dynamic programming algorithm (Lang, 1974), which can be expressed using the abstraction of Eq. (4). See Appendix B for details.

## 4 COMPARISON OF DIFFERENTIABLE STACKS

Here we make the novel insight that the superposition stack is a special case of the dVPDA. If we unroll the superposition stack's equation for $\boldsymbol{r}_t$, we see that it is a summation that enumerates all possible sequences of actions:

$$\boldsymbol{r}_t = \sum_{\pi \rightsquigarrow t} \psi(\pi) \, \boldsymbol{v}(\pi) \qquad (13)$$

where $\pi$ is a run of a VPDA with $|Q| = |\Gamma| = 1$, and $\pi \rightsquigarrow t$ means that $\pi$ ends at timestep $t$. This means the superposition stack is a special case of the dVPDA with $|Q| = |\Gamma| = 1$, normalized transition weights, and $\boldsymbol{v}_0 = \boldsymbol{0}$. So, like the dVPDA, the superposition stack is a soft-selection over VPDA runs, but for a weak VPDA where nondeterministic branching can only be directed by the choice to push, pop, or do nothing at each step, and not by the state machine or stack contents.

This explains differences in the range of tasks these two stacks can solve, which we will observe in Section 6. Both stacks are **real-time**, meaning that all PDA transitions scan exactly one input symbol. The superposition stack resembles **real-time deterministic PDAs (DPDAs)**, which recognize only a subset of CFLs called **real-time deterministic CFLs (DCFLs)** (Ginsburg & Greibach, 1966; Igarashi, 1985). In contrast, nondeterministic PDAs recognize all CFLs even when they are real-time (Greibach, 1965) and in the normal form of Eqs. (9) to (11) (DuSell & Chiang, 2023). Therefore dVPDAs, and transformers that contain them, can recognize all CFLs.

## 5 METHOD

We construct a stack attention layer by replacing multi-head SDPA with a differentiable stack. Transformers consist of multiple **layers**, each of which contains multiple **sublayers**. Every sublayer includes layer normalization, dropout, and residual connections. In a transformer encoder, each layer consists of a self-attention sublayer followed by a feedforward sublayer. Similarly, in a transformer decoder, each layer consists of a self-attention sublayer, a cross-attention sublayer that attends to the output of an encoder, and a feedforward sublayer, in that order. Let $\boldsymbol{x}_t \in \mathbb{R}^{d_{\text{model}}}$ and $\boldsymbol{y}_t \in \mathbb{R}^{d_{\text{model}}}$ be the input and output, respectively, of a sublayer at timestep $t$. A sublayer is implemented as

$$\boldsymbol{x}'_t = \text{LAYERNORM}(\boldsymbol{x}_t) \tag{14}$$

$$\boldsymbol{y}'_t = \text{SUBLAYER}(t, (\boldsymbol{x}'_1, \ldots, \boldsymbol{x}'_n)) \tag{15}$$

$$\boldsymbol{y}_t = \boldsymbol{x}_t + \text{DROPOUT}(\boldsymbol{y}'_t). \tag{16}$$

Here, SUBLAYER is called the **sublayer function**, and it customizes the behavior of the sublayer. In a standard SDPA layer, the sublayer function is multi-head SDPA (Section 3.1). LAYERNORM is **layer normalization** (Ba et al., 2016). We use pre-norm instead of post-norm (Wang et al., 2019a; Nguyen & Salazar, 2019). Layer normalization is also applied to the output of the last layer.

Like SDPA, the superposition stack and dVPDA produce a weighted sum of input vectors (Section 4). Based on this insight, we adapt each differentiable stack into an attention sublayer by using it as the sublayer function instead of multi-head SDPA. We linearly transform each sublayer function input $\boldsymbol{x}'_t \in \mathbb{R}^{d_{\text{model}}}$ to $\boldsymbol{v}_t \in \mathbb{R}^m$, using the logistic function to ensure elements are in $(0, 1)$.

$$\boldsymbol{v}_t = \sigma(\boldsymbol{W}_{\text{v}} \boldsymbol{x}'_t) \tag{17}$$

This is similar to the way SDPA generates values (cf. Eq. (1)), and like SDPA, stack attention performs a soft-selection over them. We also linearly transform $\boldsymbol{x}'_t$ to $\boldsymbol{a}_t \in \mathbb{R}^{d_a}$, analogously to the way SDPA generates queries (cf. Eq. (1) again). For the superposition stack, we use softmax to ensure $\boldsymbol{a}_t$ is a probability distribution.

$$\boldsymbol{a}_t = \text{softmax}(\boldsymbol{W}_{\text{a}} \boldsymbol{x}'_t) \tag{18}$$

For the dVPDA, we ensure weights are non-negative using $\exp$. (In implementation, we actually calculate run weights in log space to avoid overflow, and the $\exp$ is merely implicit.)

$$\boldsymbol{a}_t = \exp(\boldsymbol{W}_{\text{a}} \boldsymbol{x}'_t) \tag{19}$$

We compute $\boldsymbol{r}_t$ from $\boldsymbol{a}_1, \ldots, \boldsymbol{a}_t$ and $\boldsymbol{v}_1, \ldots, \boldsymbol{v}_t$ according to Eq. (4), using STACK and READING for one of the two differentiable stacks. We call stack attention based on the superposition stack **superposition stack attention**, and we call stack attention based on the dVPDA **nondeterministic stack attention**. We linearly transform each $\boldsymbol{r}_t \in \mathbb{R}^{d_r}$ to get the sublayer function output $\boldsymbol{y}'_t \in \mathbb{R}^{d_{\text{model}}}$ (cf. Eq. (3)).

$$\boldsymbol{y}'_t = \boldsymbol{W}_{\text{y}} \boldsymbol{r}_t \tag{20}$$

Note that because the differentiable stack does not attend to future timesteps, we do not need to apply any masking to enforce causality. We illustrate our stack attention architecture in Fig. 1. To characterize the practical speed of stack attention on GPUs, we show its parallel time complexity in Table 1. See Appendix C for more details on time and space complexity.

$$\text{output } \{ \qquad \boldsymbol{y}_{t-1} \qquad\qquad \boldsymbol{y}_t \qquad\qquad \boldsymbol{y}_{t+1}$$

Figure 1: Conceptual diagram of a stack attention sublayer, unrolled across a portion of time. Dotted arrows indicate linear transformations, and dashed arrows indicate residual connections.

Table 1: Parallel time complexity of the types of attention studied in this paper, as a function of sequence length $n$. "Implemented" shows the complexity of each implementation used in this paper. Nondeterministic stack attention could be further parallelized in a manner similar to parallel CKY (Kosaraju, 1975), which is impossible in RNNs ("Parallel CKY"). Since CFL recognition is in $\text{NC}^2$ (Ruzzo, 1981), theoretically this could be lowered even further to $O((\log n)^2)$ ("Theoretical").

| Attention | Implemented | Parallel CKY | Theoretical |
|---|---|---|---|
| SDPA | $O(\log n)$ | – | – |
| Superposition | $O(n)$ | – | $O((\log n)^2)$ |
| Nondeterministic | $O(n^2)$ | $O(n \log n)$ | $O((\log n)^2)$ |

## 6  CONTEXT-FREE LANGUAGES

In this section, we test the performance of transformers with stack attention as language models on the same five CFL tasks from DuSell & Chiang (2020; 2022). Let $w^R$ denote the reverse of $w$.

$\boldsymbol{w\#w^R}$  The language $\{w\#w^R \mid w \in \{0,1\}^*\}$.

$\boldsymbol{ww^R}$  The language $\{ww^R \mid w \in \{0,1\}^*\}$.

$\boldsymbol{wa^pw^R}$  Like $ww^R$, but with a higher tendency to have a long stretch of the same symbol repeated in the middle. Strings are of the form $wa^pw^R$, where $w \in \{0,1\}^*$, $a \in \{0,1\}$, and $p \geq 0$.

**Dyck**  The language of strings with two types of balanced brackets.

**Hardest CFL**  A highly ambiguous, CFL-complete language with maximal parsing difficulty (Greibach, 1973). See DuSell & Chiang (2020) for details.

The $w\#w^R$ and Dyck languages are real-time DCFLs, whereas $ww^R$, $wa^pw^R$, and Hardest CFL are nondeterministic CFLs. We use the same sampling procedure as DuSell & Chiang (2020) to generate datasets for each task. Every time we train a model, we randomly sample a training set of 10k examples and a validation set of 1k examples, both with lengths in the range $[40, 80]$. For each task, we sample a test set with string lengths varying from 40 to 100, with 100 examples per length. The test sets remain the same across all runs. Following DuSell & Chiang (2022), we evaluate models using the **cross-entropy difference** between the distribution the model learns and the true distribution the data is sampled from. Units are nats, lower is better, and 0 is optimal.

We compare transformers with SDPA (Tf), superposition stack attention (Tf+Sup), and nondeterministic stack attention (Tf+Nd), as well as their stack-augmented LSTM (Hochreiter & Schmidhuber, 1997) counterparts (LSTM+Nd is what DuSell & Chiang (2023) call the VRNS-RNN). All transformers have 5 layers. For transformers with stack attention, in the third (middle) layer, we replace the SDPA sublayer with the corresponding stack attention sublayer. Although we could have used stack attention in all layers, only one is necessary for recognizing CFLs, and multiple stack attention layers would be computationally costly. Using stack attention in the middle layer is a compromise between placing it at the beginning or end of the transformer.

We have adjusted model sizes so that their parameter counts satisfy the following constraints on the Hardest CFL: (1) each type of stack has fewer parameters than the preceding types (none > Sup >

Table 2: Results of training transformers with different attention mechanisms as language models on the Penn Treebank benchmark of Dyer et al. (2016), measured with perplexity. All results are the best of 20 random restarts, selected by validation cross-entropy. Despite having the fewest parameters, our nondeterministic stack attention (Tf+Nd) has the lowest perplexity.

| Model | Params. | Val. $\downarrow$ | Test $\downarrow$ |
|---|---|---|---|
| Tf | 10,051,072 | 115.11 | 92.84 |
| Tf+Sup (Ours) | 10,050,304 | 122.94 | 98.67 |
| Tf+Nd (Ours) | 9,861,898 | **110.59** | **88.54** |

Nd); and (2) each transformer has fewer parameters than its LSTM counterpart. For transformers, the feedforward hidden layer size is $2 \cdot d_{\mathrm{model}}$. SDPA layers are causally masked and have 4 heads. For Tf, we use $d_{\mathrm{model}} = 32$. For Tf+Sup, we use $d_{\mathrm{model}} = m = 32$. For Tf+Nd, we use $d_{\mathrm{model}} = 28$, $m = 5$, and the same values for $|Q|$ and $|\Gamma|$ as DuSell & Chiang (2022) ($|Q| = 2$, $|\Gamma| = 3$ for $w\#w^R$, $ww^R$, and Dyck; $|Q| = 3$, $|\Gamma| = 3$ for $wa^pw^R$ and Hardest CFL). For LSTM, we use one layer with 100 hidden units. For LSTM+Sup, we use 93 hidden units and $m = 10$. For LSTM+Nd, we use 64 hidden units and the same dVPDA sizes as Tf+Nd. See Appendix D for more details.

For each language and architecture, we train 10 models and report results for the model with the best validation performance (Fig. 2). Tf+Sup outperforms Tf on all tasks except Hardest CFL. Tf+Nd achieves strong results across all tasks, except out-of-distribution lengths on DCFLs ($w\#w^R$ and Dyck). Notably, Tf+Nd outperforms all models on Hardest CFL for in-distribution lengths despite having the fewest parameters. It also outperforms Tf and Tf+Sup on all nondeterministic CFLs ($ww^R$, $wa^pw^R$, Hardest CFL). Although multi-head SDPA can express multiple interpretations of an input at once, it represents only a fixed number, whereas Tf+Nd sums over an exponential number of VPDA runs. The poor performance of Tf and even Tf+Sup on $ww^R$, an extremely simple nondeterministic task, highlights the importance of nondeterminism.

How do transformers compare to their LSTM counterparts? For all tasks except Dyck, LSTM outperforms Tf, and transformers have serious length generalization issues. This may have to do with the linear bias in the SDPA layers, and different positional encodings might also alleviate this (we use sinusoidal encodings). However, Tf+Nd appears to alleviate this problem on nondeterministic CFLs, but it still overfits to the training length distribution on the DCFLs. LSTM+Sup generally outperforms Tf+Sup. LSTM+Nd has better length generalization than Tf+Nd and outperforms on $ww^R$, but Tf+Nd has better in-distribution performance on Hardest CFL.

Qualitatively, we find that Tf+Sup learns easily interpretable action patterns on the real-time DCFLs. On $w\#w^R$, it learns to push before $\#$ and pop afterwards. On Dyck, it learns to push after reading opening brackets and pop after reading closing brackets. See Appendix E for details.

## 7 NATURAL LANGUAGE MODELING

In this section, we test transformers with and without stack attention on a natural language modeling benchmark. We use the Penn Treebank (Marcus et al., 1994) as preprocessed by Dyer et al. (2016). We train the models to learn a probability distribution over single sentences, without context from past sentences. Each transformer has 5 encoder layers, with stack attention inserted in the same way as Section 6. We use $d_{\mathrm{model}} = 256$ and a feedforward hidden layer size of 1024. SDPA sublayers are causally masked and have 8 heads. For superposition stack attention (Tf+Sup), we set $m = 511$. For nondeterministic stack attention, we set $|Q| = 3$, $|\Gamma| = 3$, and $m = 10$. These settings ensure that the number of parameters in each model is less than that of the preceding baselines. Note that these models are relatively small; for comparison, the Transformer-XL language model of Sartran et al. (2022) has 12M parameters. See Appendix F for more details, including computational cost.

For each architecture, we train 20 models and report results for the model with the lowest validation cross-entropy (Table 2). We see that Tf+Sup does not outperform the Tf baseline, but Tf+Nd outperforms both, despite having the fewest parameters. This suggests that nondeterministic stack attention is more efficient at encoding language under a constrained parameter budget.

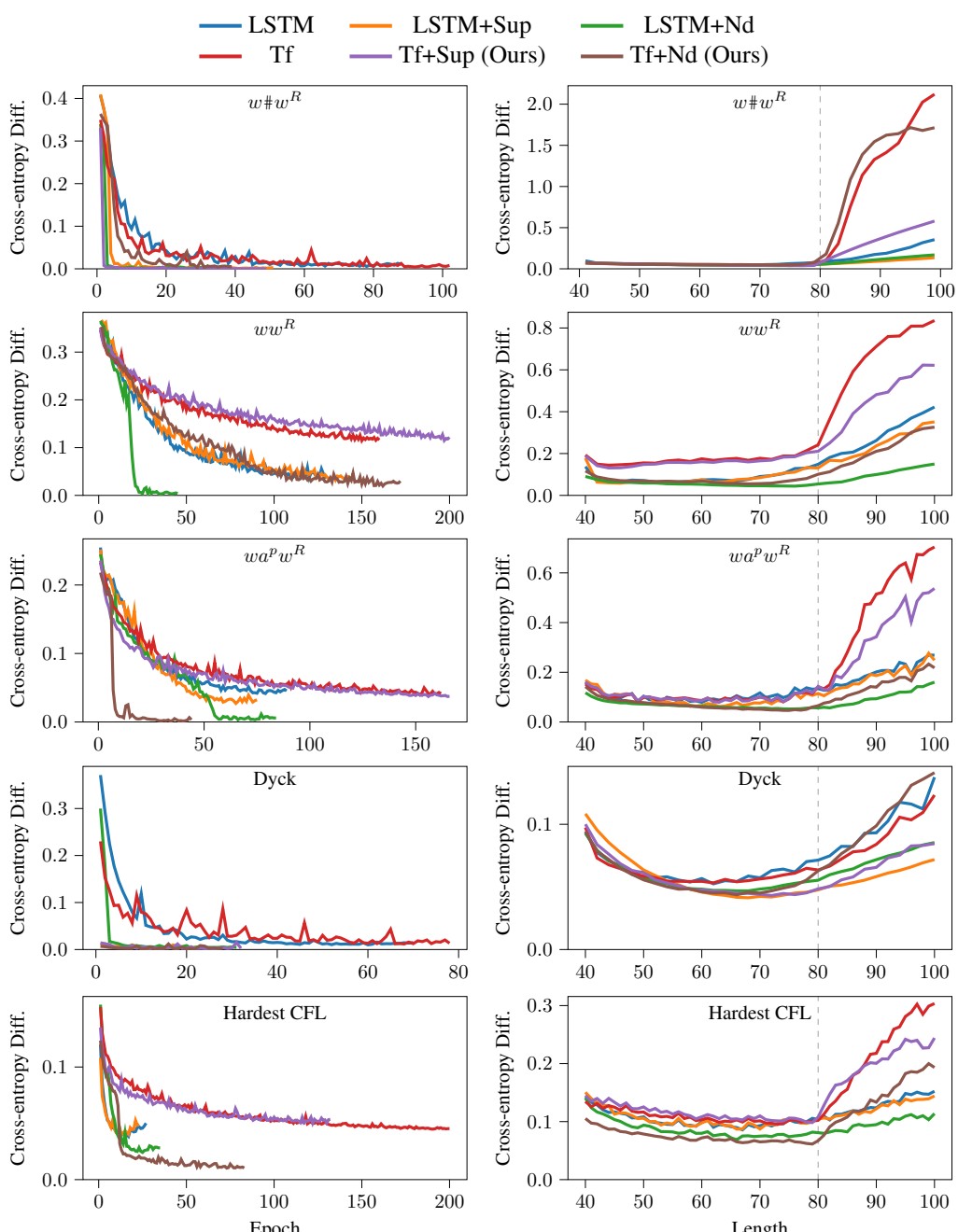

Figure 2: Language modeling results on context-free languages, comparing transformers with and without stack attention, as well as their LSTM counterparts. Left: Cross-entropy difference ($\downarrow$) in nats between model and source distribution on the validation set, as a function of training time. Lines are the best of 10 runs, selected by validation cross-entropy difference. Right: Cross-entropy difference ($\downarrow$) on the test set, binned by string length. The dashed line indicates the longest length in the training set. See Table 5 for model parameter counts. Nondeterministic stack attention (Tf+Nd) outperforms standard attention (Tf) on $ww^R$, $wa^pw^R$, and Hardest CFL; and it achieves the best in-distribution performance on Hardest CFL despite having the fewest parameters.

Table 3: Machine translation results for models trained on a subset of 100k examples from the Europarl v7 de-en corpus. Models are organized into three tiers by number of parameters. All results are the best of 5 runs, selected by decoder cross-entropy on the validation set. We report decoder perplexity on the validation set (newstest2016) and BLEU on the test set (newstest2017).

| Model | $d_{\mathrm{model}}$ | Params. | Val. Perp. $\downarrow$ | Test BLEU $\uparrow$ |
|---|---|---|---|---|
| Tf | 160 | 4,595,200 | 12.52 | **12.21** |
| Tf+Sup (Ours) | 160 | 4,492,480 | **11.94** | 12.03 |
| Tf+Nd (Ours) | 160 | 4,465,610 | 13.00 | 11.86 |
| Tf | 240 | 9,580,800 | 12.54 | 12.11 |
| Tf+Sup (Ours) | 240 | 9,349,920 | **11.53** | **12.81** |
| Tf+Nd (Ours) | 240 | 9,232,810 | 12.46 | 11.50 |
| Tf | 360 | 20,419,200 | **11.80** | **12.69** |
| Tf+Sup (Ours) | 360 | 19,900,080 | 12.03 | 12.03 |
| Tf+Nd (Ours) | 360 | 19,551,610 | 12.54 | 11.74 |

## 8 MACHINE TRANSLATION

In this section, we test transformers with stack attention on a small machine translation task, using stack attention in both the encoder and decoder. We use a subset of the German-English dataset from Europarl v7 (Koehn, 2005), simulating a low-resource scenario. We use the news translation tasks from WMT 16 (newstest2016) and WMT 17 (newstest2017) as validation and test data, respectively. To accommodate the computational cost of nondeterministic stack attention, we limit the training, validation, and test sets to examples where both the source and target side have no more than 150 characters, and we randomly subsample 100k examples from the training set. We tokenize the data into a vocabulary of 32k tokens using BPE (Sennrich et al., 2016).

We use 5 layers in both the encoder and decoder, swapping out SDPA with the same kind of stack attention in the third layer of each. We continue to use SDPA as the cross-attention mechanism. In order to examine model performance as a function of parameter count, we vary $d_{\mathrm{model}}$ to test models with three size tiers, roughly doubling the number of parameters in each successive tier, and ensuring that the number of parameters in each model is less than that of the preceding baselines. We use 8 heads in each SDPA sublayer and set the feedforward hidden layer size to $4 \cdot d_{\mathrm{model}}$. For superposition stack attention (Tf+Sup), we set $m = d_{\mathrm{model}}$. For nondeterministic stack attention (Tf+Nd), we set $|Q| = 3$, $|\Gamma| = 3$, and $m = 5$. See Appendix G for more details.

For each tier and architecture, we train 5 models and report results for the model with the lowest decoder cross-entropy on the validation set (Table 3). Stack attention does not appear to improve translation quality over Tf, although Tf+Sup performs best on the middle tier. Why does stack attention not help consistently? Perhaps natural language does not actually contain many deeply-nested hierarchies (especially on sentences limited to 150 characters), so adding a latent model of syntax does not provide a benefit over the baseline transformer, as suggested by Yao et al. (2021). However, Section 7 seems to indicate that this is not the case, at least for language modeling. Perhaps language *is* hierarchical, but baseline transformers of this size are already able to learn syntax just as effectively as those with stack attention. Or, perhaps stack attention *can* provide an advantage over baseline transformers, but the training procedure simply does not find parameter settings that do so.

## 9 CONCLUSION

We showed that two types of differentiable stack, the superposition stack (Joulin & Mikolov, 2015) and a nondeterministic generalization of it (DuSell & Chiang, 2023), can be incorporated into transformers as an attention mechanism. On CFL language modeling tasks, we showed that nondeterministic stack attention improves upon the standard transformer architecture, and even improves upon its RNN counterpart on the Hardest CFL, a challenging ambiguous language. We also showed that nondeterministic stack attention improves perplexity on a language modeling benchmark. We believe stack attention is an important development in unsupervised syntax-oriented transformers.

REPRODUCIBILITY STATEMENT

To facilitate reproducibility, we have publicly released all code we used to download and preprocess datasets, run our experiments, and generate the figures and tables in this paper. During both development and experimentation, we ran our code in containers to simplify the replication of our software environment. Our code includes the original Docker image definition we used, as well as the shell commands we used for each experiment, figure, and table. We have thoroughly documented our experimental settings in Sections 6 to 8 and Appendices D, F and G.

Although we cannot distribute the Penn Treebank dataset used in Section 7, we have included a script that generates the same preprocessed files used by Dyer et al. (2016) and Sartran et al. (2022) from the raw Penn Treebank distribution files, so that anyone with a license for the Penn Treebank can reproduce them. To our knowledge, such a script was not available previously.

We used publicly available datasets for the machine translation experiments in Section 8.

ACKNOWLEDGEMENTS

We thank Laurent Sartran for providing us with the preprocessed Penn Treebank dataset used by Dyer et al. (2016) and Sartran et al. (2022), and for answering our questions. We thank Ken Sible for helpful discussion about the implementation of our machine translation system. We thank Darcey Riley, Aarohi Srivastava, Stephen Bothwell, and Ken Sible for their feedback on a draft of this paper. We thank Alex Warstadt and Chihiro Taguchi for pointing us to some of the cited work. We thank the Center for Research Computing at the University of Notre Dame for providing the computing infrastructure used for our experiments. Finally, we thank the anonymous reviewers for their invaluable discussion and feedback. This material is partially based upon work supported by the National Science Foundation under Grant No. CCF-2019291.

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

## A   DETAILS OF PUSHDOWN AUTOMATA

Here, we define pushdown automata in more detail than in Section 3.2.2.

**Definition 1.** A *pushdown automaton (PDA)* is a tuple $(Q, \Sigma, \Gamma, \delta, q_0, F)$, where

- $Q$ is a finite set of **states**,
- $\Sigma$ is a finite **input alphabet**,
- $\Gamma$ is a finite **stack alphabet**,
- $\delta \subseteq Q \times (\Sigma \cup \{\varepsilon\}) \times \Gamma^* \times Q \times \Gamma^*$ is the **transition function**,
- $q_0$ is the **start state**, and
- $F \subseteq Q$ is the set of **accept states**.

Let $w = w_1 \cdots w_n \in \Sigma^n$ be a string of input symbols. At any given time, having read up to input position $i$, a PDA is in a state $q \in Q$ and has a stack $\beta \in \Gamma^*$ (denoted in bottom-to-top order). We encapsulate this as a **configuration** $(i, q, \beta)$. The initial configuration is always $(0, q_0, \perp)$, where $\perp \in \Gamma$ is a designated bottom symbol. If $(q, a, u, r, v) \in \delta$, we say the PDA has transition $q, u \xrightarrow{a} r, v$, which signifies that if the PDA is in state $q$ and has $u$ on top of the stack, and $a$ can be scanned from the input, then it goes to state $r$, pops $u$, and pushes $v$.

A **run** is a sequence of transitions in $\delta$, linked by configurations, starting with the initial configuration. Since the PDA is nondeterministic, multiple transitions may ambiguously apply to the same configuration, generating multiple runs per input string. We write $\pi \rightsquigarrow i, q, x$ to indicate that run $\pi$ ends in configuration $(i, q, \beta x)$ for some $\beta \in \Gamma^*, x \in \Gamma$. We say the PDA **accepts** string $w$ if there is a run $\pi \rightsquigarrow n, f, \perp$ for some $f \in F$, and **rejects** it otherwise.

## B   IMPLEMENTATION DETAILS OF THE DVPDA

Here, we provide details of implementing the dVPDA, which consists of computing $\boldsymbol{r}_0, \ldots, \boldsymbol{r}_n$ (Eq. (12)) given $\boldsymbol{\Delta}_1, \ldots, \boldsymbol{\Delta}_n$ and $\boldsymbol{v}_0, \ldots, \boldsymbol{v}_n$, where $n$ is the length of the input sequence. For an explanation of how this implementation is derived, see DuSell & Chiang (2020; 2023).

We maintain three main data structures:

- A tensor $\gamma$ of size $(n+1) \times (n+1) \times |Q| \times |\Gamma| \times |Q| \times |\Gamma|$ called the **inner weights**. We write elements as $\gamma[i \rightarrow t][q, x \rightarrow r, y]$, where $-1 \leq i \leq n-1, 0 \leq t \leq n, q, r \in Q$, and $x, y \in \Gamma$.
- A tensor $\boldsymbol{\zeta}$ of size $(n+1) \times (n+1) \times |Q| \times |\Gamma| \times |Q| \times |\Gamma| \times m$ called the **vector inner weights**. We write elements, which are members of $\mathbb{R}^m$, as $\boldsymbol{\zeta}[i \rightarrow t][q, x \rightarrow r, y]$, similarly to $\gamma$.
- A tensor $\alpha$ of size $(n+2) \times |Q| \times |\Gamma|$ called the **forward weights**. We write elements as $\alpha[t][r, y]$, where $-1 \leq t \leq n, r \in Q$, and $y \in \Gamma$.

Let $\mathbb{I}[\phi]$ denote the indicator function, which is 1 if proposition $\phi$ is true and 0 otherwise. We initialize the tensors as follows.

$$\gamma[-1 \rightarrow 0][q, x \rightarrow r, y] = \mathbb{I}[q = q_0 \wedge x = \perp \wedge r = q_0 \wedge y = \perp] \tag{21}$$

$$\boldsymbol{\zeta}[-1 \rightarrow 0][q, x \rightarrow r, y] = \mathbb{I}[q = q_0 \wedge x = \perp \wedge r = q_0 \wedge y = \perp] \, \boldsymbol{v}_0 \tag{22}$$

$$\alpha[-1][r, y] = \mathbb{I}[r = q_0 \wedge y = \perp] \tag{23}$$

For $1 \le t \le n$ and $-1 \le i \le t - 1$,

$$
\begin{aligned}
\gamma[i \to t][q, x \to r, y] = \; & \mathbb{I}[i = t - 1] \, \boldsymbol{\Delta}_t[q, x \to r, xy] && \text{push} \\
& + \sum_{s \in Q, z \in \Gamma} \gamma[i \to t{-}1][q, x \to s, z] \, \boldsymbol{\Delta}_t[s, z \to r, y] && \text{repl.} \\
& + \sum_{k=i+1}^{t-2} \sum_{u \in Q} \gamma[i \to k][q, x \to u, y] \, \gamma'[k \to t][u, y \to r]. && \text{pop}
\end{aligned}
\tag{24}
$$

Here, $\gamma'$ is an auxiliary tensor of size $(t - 1) \times |Q| \times |\Gamma| \times |Q|$ that is computed once for each $t$. Elements are written as $\gamma'[k \to t][u, y \to r]$, where $0 \le k \le t - 1$, $u, r \in Q$, and $y \in \Gamma$. For $0 \le k \le t - 2$,

$$
\gamma'[k \to t][u, y \to r] = \sum_{s \in Q, z \in \Gamma} \gamma[k \to t{-}1][u, y \to s, z] \, \boldsymbol{\Delta}_t[s, z \to r, \varepsilon].
\tag{25}
$$

For $1 \le t \le n$ and $-1 \le i \le t - 1$,

$$
\begin{aligned}
\boldsymbol{\zeta}[i \to t][q, x \to r, y] = \; & \mathbb{I}[i = t - 1] \, \boldsymbol{\Delta}_t[q, x \to r, xy] \, \boldsymbol{v}_t && \text{push} \\
& + \sum_{s \in Q, z \in \Gamma} \boldsymbol{\zeta}[i \to t{-}1][q, x \to s, z] \, \boldsymbol{\Delta}_t[s, z \to r, y] && \text{repl.} \\
& + \sum_{k=i+1}^{t-2} \sum_{u \in Q} \boldsymbol{\zeta}[i \to k][q, x \to u, y] \, \gamma'[k \to t][u, y \to r]. && \text{pop}
\end{aligned}
\tag{26}
$$

For $0 \le t \le n$,

$$
\alpha[t][r, y] = \sum_{i=-1}^{t-1} \sum_{q \in Q, x \in \Gamma} \alpha[i][q, x] \, \gamma[i \to t][q, x \to r, y].
\tag{27}
$$

For each $t$, we compute a tensor $\boldsymbol{\eta}_t$ of size $|Q| \times |\Gamma| \times m$. We write elements, which are members of $\mathbb{R}^m$, as $\boldsymbol{\eta}_t[r, y]$, where $r \in Q$ and $y \in \Gamma$.

$$
\boldsymbol{\eta}_t[r, y] = \sum_{i=-1}^{t-1} \sum_{q \in Q, x \in \Gamma} \alpha[i][q, x] \, \boldsymbol{\zeta}[i \to t][q, x \to r, y]
\tag{28}
$$

Finally, we compute the stack reading $\boldsymbol{r}_t$, implementing Eq. (12), as follows.

$$
\boldsymbol{r}_t[r, y] = \frac{\boldsymbol{\eta}_t[r, y]}{\sum_{r' \in Q, y' \in \Gamma} \alpha[t][r', y']}
\tag{29}
$$

To prevent underflow and overflow, we compute $\gamma$, $\boldsymbol{\zeta}$, $\alpha$, $\gamma'$, and $\boldsymbol{\eta}$ entirely in log space.

How do these equations fit with Eq. (4)? Note that $\gamma$, $\boldsymbol{\zeta}$, and $\alpha$ can be computed incrementally in order of increasing $t$. To achieve the right time and space complexity, it is important to pre-allocate $\gamma$, $\boldsymbol{\zeta}$, and $\alpha$ and update them in-place. We can represent the stack state $\mathcal{S}_t$ as $(t, \gamma, \boldsymbol{\zeta}, \alpha)$. Then $\text{STACK}((t - 1, \gamma, \boldsymbol{\zeta}, \alpha), \boldsymbol{\Delta}_t, \boldsymbol{v}_t)$ computes all values up to timestep $t$, assuming that all values up to $t - 1$ have been computed, and increments $t - 1$ to $t$. We let $\text{READING}((t, \gamma, \boldsymbol{\zeta}, \alpha))$ implement Eqs. (28) and (29).

## C    TIME AND SPACE COMPLEXITY OF STACK ATTENTION

In Table 4, we show the serial time and space complexity of the three types of attention studied in this paper: SDPA, superposition stack attention (Sup), and nondeterministic stack attention (Nd). In the case of superposition stack attention, if only the forward pass is needed, past stacks can be discarded at each timestep, reducing the space complexity by a factor of $O(n)$.

Table 4: Serial time and space complexity of the types of attention studied in this paper. "Serial Time" and "Space" are for a combined forward-backward pass; "Space (Forward Only)" is for a forward pass only. Here, $n$ is the length of the input sequence, $d_k$ is the size of the query/key vectors in SDPA, $d_v$ is the size of the value vectors in SDPA, $m$ is the size of the pushed stack vectors $\boldsymbol{v}_t$ in stack attention, $Q$ is the set of PDA states in the nondeterministic stack, and $\Gamma$ is the set of stack symbols in the nondeterministic stack. For simplicity, we disregard the linear transformations of Eqs. (1) and (17) to (20).

| Attention | Serial Time | Space | Space (Forward Only) |
|---|---|---|---|
| SDPA | $O(d_k n^2 + d_v n^2)$ | $O(n^2 + d_k n + d_v n)$ | same |
| Sup | $O(mn^2)$ | $O(mn^2)$ | $O(mn)$ |
| Nd | $O(m|Q|^3|\Gamma|^3 n^2 + m|Q|^3|\Gamma|^2 n^3)$ | $O(m|Q|^2|\Gamma|^2 n^2)$ | same |

In order to provide a better picture of the practical speed of stack attention on GPUs, in Table 1, we show the *parallel* time complexity of our current implementations of stack attention, as well as improved parallel time complexity that could be attained after applying known speedups.

Stack attention introduces a recurrence that does not exist in SDPA (see Eq. (4)), so it cannot be parallelized across the timestep dimension as easily. (Note that SDPA requires a softmax over $n$ inputs, which can be done in $O(\log n)$ parallel time.) However, stack attention in transformers does present opportunities for parallelization that do not exist in stack-augmented RNNs. In stack-augmented RNNs, the actions and pushed vector for timestep $t$ depend on the hidden state at timestep $t$. Since the hidden state computation cannot be parallelized across $t$, neither can the stack computation. On the other hand, in a transformer with stack attention, the stack actions and pushed vectors for all $t$ are computed in parallel by earlier SDPA layers and are available all at once. This means that we can use techniques that parallelize the stack computation across $t$.

In the case of nondeterministic stack attention, our current implementation, which is based on Lang's algorithm (see Appendix B), populates $\gamma$ and $\boldsymbol{\zeta}$ serially for increasing $t$, incurring a cost of $\Theta(n)$ parallel time. The update at each step is parallelized over $i$, but the summation over $k$ runs in $\Theta(k)$ parallel time. So, the overall parallel time complexity is $O(n^2)$.

We can parallelize Lang's algorithm over $t$ in a manner very similar to that of parallel CKY parsing (Kosaraju, 1975). This would require $n$ iterations that populate $\gamma$ and $\boldsymbol{\zeta}$ in increasing order of span size $t-i$. Each iteration would, in parallel over each $i$, compute the terms for all splits in the pop rule in parallel over $k$, and sum them in $O(\log k)$ parallel time. So, the overall parallel time complexity can be lowered to $O(n \log n)$. Since CFL recognition is in $\text{NC}^2$ (Ruzzo, 1981), theoretically this can be lowered even further to $O((\log n)^2)$.

In the case of superposition stack attention, each stack element depends only on a constant number of elements (three) from the previous timestep, so each stack update can be done in $O(1)$ parallel time. Our implementation performs stack updates in order of increasing $t$, so overall it runs in $O(n)$ parallel time. Since superposition stack attention is a special case of nondeterministic stack attention, it can also be theoretically lowered to $O((\log n)^2)$, and perhaps further.

GPU memory limitations are sometimes a practical challenge when using nondeterministic stack attention. For the machine translation experiments in particular, which use separate stack attention modules in both the encoder and decoder, we had to contend with memory fragmentation issues in PyTorch's CUDA memory allocator.

For examples of wall-clock runtimes and GPU memory cost, see Table 6.

## D   DETAILS OF CONTEXT-FREE LANGUAGE EXPERIMENTS

Here, we provide additional details about the models and training procedure used in Section 6.

Table 5: Parameter counts for each task and each model used in Section 6. Parameter counts differ across tasks due to differences in vocabulary size.

| Task | LSTM | LSTM+Sup | LSTM+Nd | Tf | Tf+Sup | Tf+Nd |
|---|---|---|---|---|---|---|
| $w\#w^R$ | 42,004 | 41,402 | 31,138 | 43,044 | 40,964 | 33,273 |
| $ww^R$ | 41,503 | 40,936 | 30,817 | 42,979 | 40,899 | 33,216 |
| $wa^pw^R$ | 41,503 | 40,936 | 41,482 | 42,979 | 40,899 | 36,576 |
| Dyck | 42,505 | 41,868 | 31,459 | 43,109 | 41,029 | 33,330 |
| Hardest CFL | 44,008 | 43,266 | 43,087 | 43,304 | 41,224 | 36,861 |

### D.1 MODELS

For LSTMs, inputs are encoded as one-hot vectors. We use PyTorch's (Paszke et al., 2019) LSTM implementation, which by default includes redundant bias parameters $b_{\mathrm{hi}}$, $b_{\mathrm{hf}}$, $b_{\mathrm{hg}}$, and $b_{\mathrm{ho}}$. We remove these parameters in our experiments and from our reported parameter counts. We set the initial hidden state and memory cell to $\mathbf{0}$.

For transformers, we use PyTorch's transformer layer implementation. Like Vaswani et al. (2017), we map inputs to vectors of size $d_{\mathrm{model}}$ with a scaled embedding layer and apply sinusoidal positional encodings. We do not tie input and output embeddings. The first input token to each transformer is always BOS. We train all models to predict EOS as the last output token. We map the outputs of the final layer to logits for predicting the next token via a learned affine transformation. We use a dropout rate of 0.1 throughout the transformer; PyTorch applies it in the same places as Vaswani et al. (2017), and additionally to the hidden units of feedforward sublayers and the attention probabilities of SDPA ($z_t$ in Eq. (2)). We apply dropout to the hidden units of the feedforward sublayers of stack attention layers as well.

We show parameter counts for all models on all tasks in Table 5.

### D.2 INITIALIZATION

We initialize all fully-connected layers with Xavier uniform initialization (Glorot & Bengio, 2010) except for the recurrent LSTM layer and all fully-connected layers in SDPA layers. For layer norm, we initialize all weights to 1 and all biases to 0. We initialize all other parameters by sampling uniformly from $[-0.1, 0.1]$.

### D.3 TRAINING

We use minibatches of size 10. We generate batches once before training; to generate a batch, we first select a length and then sample 10 strings of that length. We randomly shuffle batches before each epoch. We train each model by minimizing its cross-entropy (summed over the timestep dimension of each batch) on the training set, and we use per-symbol cross-entropy on the validation set as the early stopping criterion. We optimize parameters with Adam (Kingma & Ba, 2015), and we randomly sample the initial learning rate from a log-uniform distribution over $[5 \times 10^{-4}, 1 \times 10^{-2}]$. We clip gradients with a threshold of 5 using $L^2$ norm rescaling. We multiply the learning rate by 0.9 after 5 epochs of no improvement in cross-entropy on the validation set, and we stop early after 10 epochs of no improvement. We train for a maximum of 200 epochs.

## E ANALYSIS OF STACK ATTENTION

In Figs. 3 and 4, we show heatmaps of the stack actions learned by Tf+Sup on the real-time DCFL tasks. These actions are from the same models whose results are shown in Fig. 2.

## F DETAILS OF NATURAL LANGUAGE MODELING EXPERIMENTS

Here, we provide additional details about the models and training procedure used in Section 7.

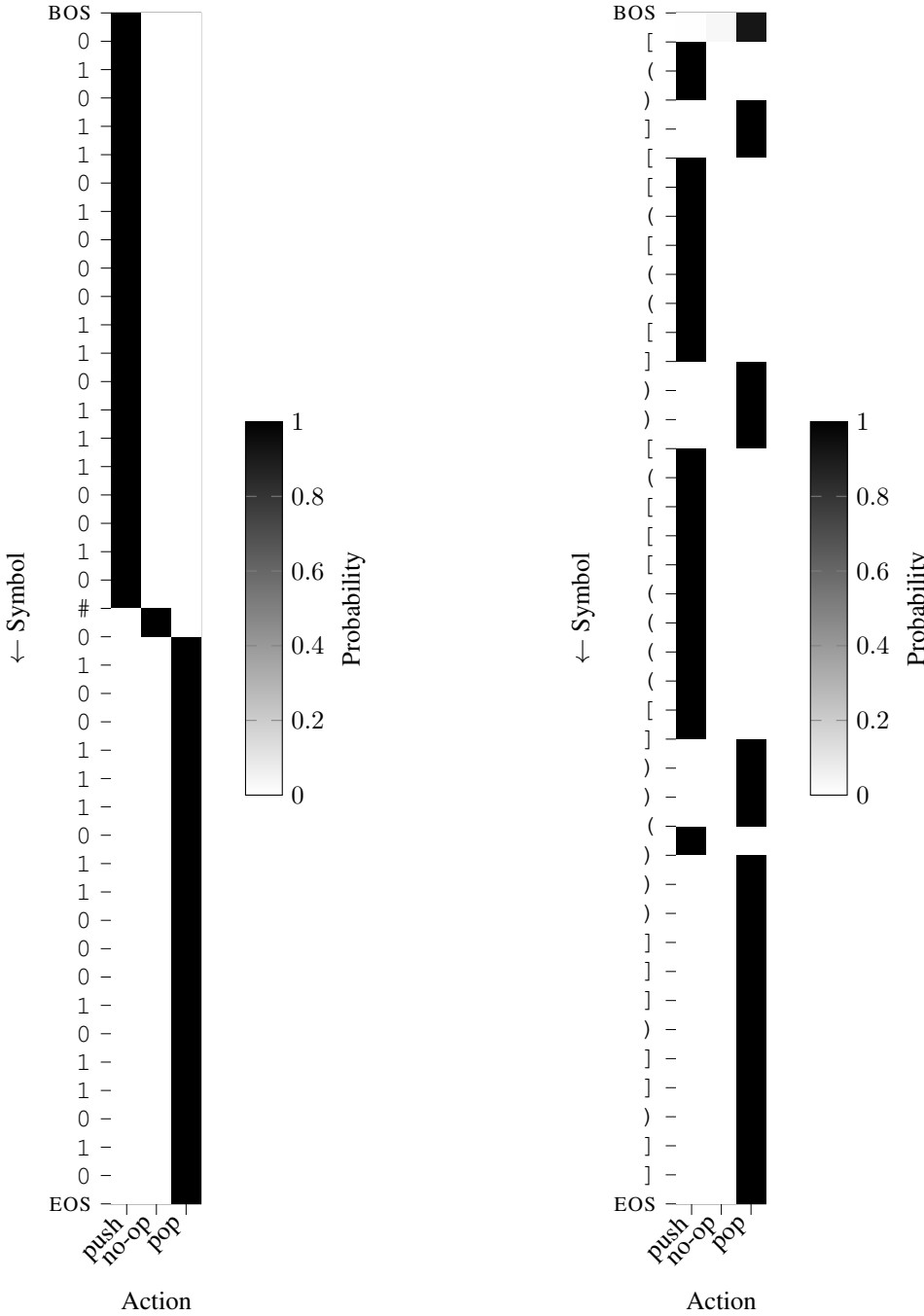

Figure 3: Visualization of superposition stack attention on a string in $w\#w^R$. As expected, the model learns to push all symbols before $\#$, do nothing when reading $\#$, and pop all symbols after $\#$.

Figure 4: Visualization of superposition stack attention on a string in the Dyck language. As expected, the model learns to push opening brackets and pop when reading closing brackets.

Table 6: Computational cost of training each architecture on the PTB language modeling task when run on an NVIDIA TITAN Xp GPU.

| Model | Examples/s | Minutes/Epoch | GPU Memory |
|---|---|---|---|
| Tf | 859 | 0.8 | 394 MB |
| Tf+Sup | 345 | 1.9 | 397 MB |
| Tf+Nd | 27 | 24.3 | 1.91 GB |

## F.1 MODELS

The first input token to each transformer is always BOS, and we train it to predict EOS as the last output token. We use a dropout rate of 0.1 in all the same places as in Appendix D.1. We use sinusoidal positional encodings. We do tie the embeddings of the input and output layers.

## F.2 INITIALIZATION AND TRAINING

We initialize parameters in the same way as Appendix D.2, except the interval used for uniform sampling is $[-0.01, 0.01]$.

For each run, we randomly sample a batch size $B$ from a uniform distribution over $[128, 512]$. For each epoch, we randomly shuffle examples and group examples of similar lengths into the same minibatch, enforcing an upper limit of $B$ tokens per batch, including padding, BOS, and EOS tokens. We clip gradients with a threshold of 5 using $L^2$ norm rescaling.

We optimize parameters using Adam. For each run, we randomly sample the initial learning rate from a log-uniform distribution over $[10^{-6}, 10^{-4}]$. We take a checkpoint every 20k examples to evaluate the model's cross-entropy on the validation set. We multiply the learning rate by 0.5 after 2 checkpoints with no improvement on the validation set, and we stop early after 4 checkpoints with no improvement. We use the checkpoint with the lowest validation cross-entropy.

We show the computational cost of training each architecture on this task in Table 6.

## G DETAILS OF MACHINE TRANSLATION EXPERIMENTS

Here, we provide additional details about the data preprocessing, models, training procedure, and decoding algorithm used in Section 8.

## G.1 DATA PREPROCESSING

We limit the training, validation, and test sets to examples where the source and target side each have no more than 150 Unicode characters after applying NFKC normalization. In the training set, we filter out training examples with empty sentences and those where the length in characters of one side is more than 4 times that of the other. After filtering, we train a tokenizer on the combined source and target sentences of the subsampled training data using the SentencePiece (Kudo & Richardson, 2018) implementation of BPE (Sennrich et al., 2016). We use a vocabulary of 32k tokens and a frequency threshold of 50.

## G.2 MODELS

We always give the encoder EOS as the last input token to allow it to detect the end of the source sequence. We always give the decoder BOS as the first input token and have it predict EOS as the last output token. SDPA is not causally masked in the encoder, but it is in the decoder. Like Vaswani et al. (2017), we tie the embeddings of the encoder input, decoder input, and decoder output layers. We use a dropout rate of 0.1 as in Appendix F.1, applying it also to the softmax probabilities of cross-attention.

### G.3 INITIALIZATION AND TRAINING

We initialize parameters in the same way as Appendix F.2. For every epoch, we randomly shuffle the training data and group examples where the combined source and target length in tokens is similar into the same minibatch. We limit the number of tokens in the source or target side of a batch to 2048 tokens, including padding, BOS, and EOS symbols. We use label smoothing with a weight of 0.1. We clip gradients with a threshold of 5 using $L^2$ norm rescaling.

We optimize parameters using Adam. We randomly sample the initial learning rate from a log-uniform distribution over $[10^{-5}, 10^{-3}]$. We take a checkpoint every 50k examples to evaluate the decoder's cross-entropy on the validation set. We multiply the learning rate by 0.5 after 2 checkpoints with no improvement on the validation set, and we stop early after 4 checkpoints with no improvement. We train for a maximum of 100 epochs. We use the checkpoint with the best validation cross-entropy.

### G.4 DECODING

We use beam search for decoding, using a beam size of 4. During beam search, we apply length normalization to the probabilities of hypotheses before selecting the top $k$ hypotheses for the next beam. We do this by dividing the log-probability of each hypothesis by the number of tokens generated by that hypothesis so far (including EOS).

