# OpenReview forum: "Stack Attention: Improving the Ability of Transformers to Model Hierarchical Patterns"
_ICLR.cc/2024/Conference — ICLR 2024 spotlight_

### Official Review · Reviewer_bkRf · 2023-10-29

**Soundness:** 3 good
**Presentation:** 3 good
**Contribution:** 3 good
**Rating:** 8
**Confidence:** 5

**Summary:**

This paper addresses the problem of the lack of modeling the hierarchical structure of input sequences in transformers. In response, a new attention mechanism, called stack attention, is proposed. The idea is to frame the sequence modeling problem as a task of running a PDA on the input vectors; that is, each time we take a vector as input, we update the stack of the PDA and produce a new vector as output. In this way, the sequence of input vectors is represented as a sequence of output vectors. The entire process is based on differentiable states and operations. Thus, this attention model operates in the same manner as those in sequence modeling, making it easy to incorporate the model into transformers. However, as a side effect, stack attention introduces recurrence into modeling, thus preventing training from being parallelized across the sequence. The stack attention models are tested on synthetic data generated by context-free grammars. Experimental results show that nondeterministic stack attention models surpass standard transformers and achieve better results than a strong nondeterministic stack RNN baseline. The stack attention models also show promising results on small-scale machine translation and language modeling tasks.

**Strengths:**

I like this work! Given that language structure is not explicitly modeled in current Transformer models, this work opens a door to a new approach to considering hierarchical patterns in modeling languages. The design of the model is simple and elegant. The experiments support the claims well.

**Weaknesses:**

I have no major concerns, but a few comments.

The state of the stack attention models at a given timestep depends on its past state. This means the models share similar drawbacks and merits with recurrent models like RNNs. Compared to the self-attention used in standard Transformers, stack attention is slower for training because it processes one token at a time, rather than parallelizing the encoding process over the entire sequence. The author states in the appendix that this could be improved using the parallel prefix sum method, but no details are presented.

A related problem is that the experiments here are small-scale. While it's fine to test the models on synthetic data for CFL tasks, the results on language modeling and machine translation aren't comparable to those in other papers. I understand that computational cost is a concern. However, to demonstrate the superiority of stack attention, it's necessary to compare it with previously reported results under the same setup.

I’m not quite satisfied that the models are motivated by handling hierarchical structure behind languages but there is no discussion on what structure is captured. A simple way to examine this is to design probing tasks to see how much syntax is modeled in stack attention and to see how the learned syntax differs from human-annotated syntax. Unsupervised learning of syntactic structures can offer new insights into modeling natural languages."

There have been previous studies on extending standard attention models to hierarchical models, such as hierarchical attention and selective attention. These should be considered baselines for comparison, either in related work or experiments.

**Questions:**

N/A

---

> ### Author Response · Authors · 2023-11-16
>
> Dear reviewer bkRf,
>
> Thank you very much for your review and your many thoughtful comments. We are glad that you liked the paper! We greatly appreciate your suggestions for improving it further.
>
> > The state of the stack attention models at a given timestep depends on its past state. This means the models share similar drawbacks and merits with recurrent models like RNNs. Compared to the self-attention used in standard Transformers, stack attention is slower for training because it processes one token at a time, rather than parallelizing the encoding process over the entire sequence. The author states in the appendix that this could be improved using the parallel prefix sum method, but no details are presented.
>
> Your comments about the recurrence introduced by the stack are exactly correct. Actually, stack attention, despite having a recurrence across the timestep dimension, can still parallelize the stack computation in a way that is impossible in an RNN under the controller-stack paradigm used by Grefenstette et al. (2015), Joulin and Mikolov (2015), DuSell and Chiang (2020), etc. In an RNN, the stack actions for timestep $t$ depend on the hidden state at timestep $t$. Since the hidden state computation cannot be parallelized across $t$, neither can the stack. However, in a transformer with stack attention, the stack actions $\mathbf{a}_t$ and pushed vectors $\mathbf{v}_t$ for all timesteps are computed in parallel and are available all at once. This means we can take advantage of algorithms that parallelize the stack computation over the timestep dimension, such as parallel CKY. (Apologies, we realized that the algorithm for parallelizing nondeterministic stack attention across the timestep dimension would resemble parallel CKY, not parallel prefix sum as stated in Appendix B. We will update Appendix B.)
>
> For nondeterministic stack attention, the algorithm would work by updating the dynamic programming table (see Appendix A) in increasing order of span size, then parallelizing the inner loops that implement the replace and pop rules. This would reduce the runtime to $O(n)$ parallel time. As for the superposition stack, each recurrent step runs in $O(1)$ parallel time under our current implementation, so it already runs in $O(n)$ parallel time.
>
> > I’m not quite satisfied that the models are motivated by handling hierarchical structure behind languages but there is no discussion on what structure is captured. A simple way to examine this is to design probing tasks to see how much syntax is modeled in stack attention and to see how the learned syntax differs from human-annotated syntax. Unsupervised learning of syntactic structures can offer new insights into modeling natural languages."
>
> Thank you for raising this point. We agree that analyzing the parses learned by stack attention would be extremely informative and would further validate our method. If we have time, we will update our response with an analysis of how the networks learn to use stack attention.
>
> > There have been previous studies on extending standard attention models to hierarchical models, such as hierarchical attention and selective attention. These should be considered baselines for comparison, either in related work or experiments.
>
> Thank you for this suggestion. To clarify, are you referring to these papers?
>
> * Hierarchical Attention Networks for Document Classification (Yang et al.)
> * Selective Attention for Context-aware Neural Machine Translation (Maruf et al.)

---

> > ### Comment · Reviewer_bkRf · 2023-11-18
> >
> > Thank you for your response, it addresses my concerns. Yes, the references you provided are exactly what I meant.

---

### Official Review · Reviewer_knG9 · 2023-11-01

**Soundness:** 3 good
**Presentation:** 3 good
**Contribution:** 2 fair
**Rating:** 6
**Confidence:** 4

**Summary:**

The authors present work on stack attention for transformers that attempts to address naturally hierarchically structured problems. They carefully present scaled dot-product attention, a core component of the transformer architecture. Next, they provide background on differentiable stacks, superposition stacks, and a non-deterministic stack -- the differentiable vector push-down automaton. Superposition stacks can be seen as a special case of this. This is used to replace scaled dot-product attention.

They explore the empirical performance of this approach on a range of tasks, including constructed languages, a small scale language modeling problem, and a small scale machine translation effort. In some settings, the Tf-Nd configuration outperforms a conventional transformer architecture. However, these gains are not huge, the datasets are small, and (in machine translation) as the dimensionality increases, the architecture no longer shows gains.

**Strengths:**

The authors carefully present their formalism, including details and relevance connections to prior work.

The description of the method stands alone reasonably well, though familiarity with related work makes the paper much more accessible.

The authors evaluate in a range of settings, from constructed language to actual use cases.

The methodology does not rely on a specific grammar or automaton structure; instead it is latent.

**Weaknesses:**

The evaluation settings are quite small by modern standards.

Gains (at least in machine translation) seem to disappear as model size increases.

Computational cost at inference seems greater according to the big-O runtimes -- is this a net win? Is it feasible to train at larger scale?

**Questions:**

What is the empirical cost of running these methods?

---

> ### Author Response · Authors · 2023-11-16
>
> Dear reviewer knG9,
>
> Thank you very much for your thoughtful review and positive comments.
>
> > The evaluation settings are quite small by modern standards.
>
> > Gains (at least in machine translation) seem to disappear as model size increases.
>
> It's true that the models and datasets are relatively small compared to some other papers, but we believe our experiments demonstrate an important point: that transformers with a latent model of syntax are more expressive than standard transformers (as shown by the results on synthetic CFLs such as the hardest CFL), and that they can be more parameter- and data-efficient than standard transformers when the data can be explained with underlying hierarchical rules (as shown by the language modeling results).
>
> > Computational cost at inference seems greater according to the big-O runtimes -- is this a net win? Is it feasible to train at larger scale?
>
> > What is the empirical cost of running these methods?
>
> We discuss the computational cost of stack attention in detail in Appendix B, including wall-clock runtimes on the PTB language modeling task and asymptotic time/space complexity. With our current implementation, large-scale training is probably within reach for superposition stack attention but not yet for nondeterministic stack attention; superposition stack attention trains at 40\% the speed of standard attention, whereas nondeterministic stack attention trains at about 3\% speed. However, we can likely improve the runtime of nondeterministic stack attention substantially by parallelizing the dVPDA computation across the timestep dimension using a strategy similar to that of parallel CKY, which would reduce it to only $O(n)$ parallel time. We can also get a further, smaller speedup by parallelizing the computation of the stack reading $\mathbf{r}_t$ from $\mathcal{S}_t$.

---

### Official Review · Reviewer_ecSr · 2023-11-04

**Soundness:** 3 good
**Presentation:** 2 fair
**Contribution:** 3 good
**Rating:** 6
**Confidence:** 4

**Summary:**

This paper presents a stack-augmented transformer to help overcome some of the challenges of ordinary self-attention in modeling nested syntactic structure.

The overall approach is to propose two different stack mechanisms: the "superposition stack" which is an extension of Joulin and Mikolov (2015), and a second non-deterministic diferentiable vector pushdown automata (dVPDA). While the details for the dVPDA are not very clear from the methods section (see Weaknesses and questions), both the dVPDA and the superposition stack provide soft-stack vector readouts at each time-step, and can be used to replace standard attention.

From results, we find:
- On formal languages such as Dyck, 5 layer transformers with the stack based attention (coming from the superposition stack) are better than ordinary 5 layer transformers (though both of these are worse than LSTM variants).
- On language modeling over the penn treebank, we find that transformers + dVPDA obtain lower perplexities than ordinary transformers (though transformers + superposition stack are much worse).
- On a 5 layer machine translation dataset, we find slight very mixed improvements in BLEU.

**Strengths:**

The motivation behind this paper is great - there are clear limitations of self-attention w.r.t. modeling syntactic patterns and here, we see a novel approach to use a stack to model such patterns.

**Weaknesses:**

- Unfortunately, I think the results are very mixed, experiments are done on very small 5 layer transformers on small datasets.
- Even for the positive results, there is no analysis of why the stack augmented model may be doing better on natural language - is it discovering good parses / something else?

-  The exposition is very confusing, and i'm a bit lost on various details. The biggest missing detail is how training is done in parallel - from Eq 8, 9 ..., 17 and Figure-1 it seems like there is a stack state that is recurrently updated, but transformer training is fully parallel, so how can state information be passed between different tokens? Is this done by basically reconstructing the previous state of the stack since all previous actions are available to the model at each time step - if so what is the FLOP hit from doing this?

**Questions:**

I'll intersperse questions (Q) with some suggestions for improving writing (S)

Introduction:

Q1: "Recent work has shown that transformers have linear rather than hierarchical..": Is this true? Murty 2023 ("Grokking") find that transformers acquire hierarchical bias when trained for long. It would be better to qualify this statement somewhat.

S1: "on a natural language modeling benchmark"  => would be better to just say "Penn TreeBank". In general, I found the last para to be very vague. It would be better to have concrete numbers and dataset names.

Related Work:

S2: Missing several keys papers (Ordered Memory, Unsupervised Tree LSTMs, RL-SPINN) and early works on incorporating stack mechanism into transformers (Das, 1993). This is just a quick list, but there is a long history of learning syntax unsupervisedly / augmenting neural models with stacks that is completely missing.

Background Section 3.2.1:

Q3: This method seems like it would run faster than the method from 3.2.2. Could the authors confirm this?

Q4: In natural language, one might want to do multiple reduce operations after emitting a word. How does this approach allow for multiple reduces at each time step?

Background Section 3.2.2:

S3: Definition 1 and the next paragraph take too much space and seem like background that could be in an appendix.

Q5: Missing detail: What is the time complexity of Eq. 17 -  Please include pseudocode here.

Q6: "Each $a_t$ is a flattening of tensor...": How was the size of the tensor $\Delta_t$ computed here?

---

> ### Author Response · Authors · 2023-11-16
>
> Dear reviewer ecSr,
>
> Thank you very much for your thoughtful review. We greatly appreciate your constructive comments and your suggestions for improving the clarity of the paper.
>
> > Unfortunately, I think the results are very mixed, experiments are done on very small 5 layer transformers on small datasets.
>
> It's true that the models and datasets are relatively small compared to some other papers, but we believe our experiments demonstrate an important point: that transformers with a latent model of syntax are more expressive than standard transformers (as shown by the results on synthetic CFLs such as the hardest CFL), and that they can be more parameter- and data-efficient than standard transformers when the data can be explained with underlying hierarchical rules (as shown by the language modeling results).
>
> > Even for the positive results, there is no analysis of why the stack augmented model may be doing better on natural language - is it discovering good parses / something else?
>
> Thank you for raising this point. We agree that analyzing the parses learned by stack attention would be extremely informative and would further validate our method. If we have time, we will update our response with an analysis of how the networks learn to use stack attention.
>
> > The biggest missing detail is how training is done in parallel ...
>
> You are correct that the stack state $\mathcal{S}_t$ is updated recurrently. In our implementation, we simply compute $\mathcal{S}_1, \mathcal{S}_2, \ldots$ serially and compute the other parts of the transformer network in parallel as usual. Consider Figure 1, which depicts a single stack attention sublayer: we first compute all of the inputs $\mathbf{x}_t$ and apply layer norm in parallel, then we compute the stack states $\mathcal{S}_t$ and their stack readings $\mathbf{r}_t$ serially, and finally we apply dropout and residual connections in parallel to compute the outputs $\mathbf{y}_t$. Although our implementation does not currently do so, it would also be possible to parallelize the computation of $\mathbf{r}_t$ from $\mathcal{S}_t$, which would result in a small speedup for nondeterministic stack attention.
>
> Despite the recurrence of $\mathcal{S}_t$, in the case of nondeterministic stack attention, it is still possible to parallelize the computation across the timestep dimension using a strategy similar to that of parallel CKY parsing, which would reduce it to $O(n)$ parallel time and likely result in a substantial speedup over our current implementation.
>
> Q1: Thank you for pointing this out; this is indeed a very relevant finding. We will update this statement in our paper. It is interesting to note that although training for longer does improve hierarchical generalization by a lot, vanilla transformers still fall 8 to 20 percent short of perfect on question formation and tense reinflection, indicating there is still room for improvement.
>
> S1: We will follow this suggestion. Thank you!
>
> S2: Thank you for these references. Incorporating stacks and latent syntax into neural networks has become a very long line of work, so we limited our citations specifically to syntax-oriented transformers (as opposed to RNNs), and RNNs that use the differentiable stacks that we use in this paper. If we have space, we will include more discussion of other work.
>
> Q3: Yes, superposition stack attention is faster than nondeterministic stack attention. For details, please see Appendix B.
>
> Q4: Superposition stack attention does not appear to have a mechanism for doing this. As discussed in Section 4, superposition stack attention is very likely less expressive than nondeterministic stack attention, which is at least weakly equivalent to any PDA that performs multiple reductions per timestep, since it can recognize all CFLs (DuSell and Chiang, 2023).
>
> S3: Thank you; we will consider ways to make this more concise while still establishing the essential notation used in the paper.
>
> Q5: The purpose of Equation 17 is to define the output of the dVPDA mathematically in a way that is concise, if inefficient. If implemented naively, it would take time exponential in $n$, because $\pi$ loops over sequences of length $n$ in the denominator. However, it is possible to implement this equation in cubic time and quadratic space using a dynamic programming algorithm called Lang's algorithm (this is the same time/space complexity as parsing algorithms for context-free grammars, such as CKY). Since Equation 17 is sufficient for understanding the dVPDA mathematically and the implementation details are somewhat complex, we put them in the appendix to save space and to reduce the cognitive burden on the reader. For the full implementation details of Equation 17, please see Appendix A, and for a discussion of its time complexity, please see Appendix B.
>
> (continued below)

---

> > ### Author Response · Authors · 2023-11-16
> >
> > (continued from above)
> >
> > Q6: Thank you, we should have explained this better. Let us write the weight of transition $q, x \xrightarrow{w_t} r, v$ as $\Delta_t[q, x \rightarrow r, v]$, where $q, r \in Q$ are states, $x \in \Gamma$ is the popped stack symbol, and $v \in \Gamma^\ast$ is the string of pushed stack symbols. The first three dimensions of $\Delta_t$, $|Q| \times |\Gamma| \times |Q|$, correspond to the variables $q, x, r$. According to Equations 14-16, the string $v$ has only three forms, where $y \in \Gamma$:
> >
> > 1. push: $xy$
> > 2. replace: $y$
> > 3. pop: $\varepsilon$
> >
> > So there are $|\Gamma| + |\Gamma| + 1$ possible choices for $v$, which is why the last dimension is $(2|\Gamma| + 1)$.

---

> > > ### Comment · Reviewer_ecSr · 2023-11-17
> > >
> > > Thanks for the clarifications! I agree that exploring new models that allow transformers to explictly model syntax unsupervisedly is a great motivation. Just a couple of followups:
> > >
> > > - Re: parallel vs recurrent, based on your explanation, it appears that only layers below the first stack attention layer are parallelizable while the rest of the model is still recurrent? I would also still like to know the FLOP hit from this recurrent computation. And additionally, could you also please confirm the addition parameters that are added in stack attention?
> > >
> > > - The superposition stack works much better than the non-deterministic stack (both in terms of speed and results on natural language data) but as you said, it is unable to make multiple reduces at each token. Does this then mean that it cannot produce natural language syntax trees?

---

> ### Author Response · Authors · 2023-11-19
>
> > Re: parallel vs recurrent, based on your explanation, it appears that only layers below the first stack attention layer are parallelizable while the rest of the model is still recurrent?
>
> The layers after stack attention remain parallelizable as well. The procedure is essentially:
>
> 1. compute all layers before stack attention in parallel to get $\mathbf{a}_t$ and $\mathbf{v}_t$ for all $t$
> 2. run stack attention serially to get $\mathbf{r}_t$ for all $t$
> 3. now that we have all $\mathbf{r}_t$, compute all $\mathbf{y}_t$ (see Figure 1) and all layers after stack attention in parallel
>
> > I would also still like to know the FLOP hit from this recurrent computation.
>
> We have not yet worked out the implementation details of the parallelized version, so at this time it's difficult to quantify the difference in FLOPs. In terms of time complexity, the current implementation runs in $O(n^2)$ parallel time, whereas the parallel CKY-style version would run in $O(n)$ parallel time.
>
> > And additionally, could you also please confirm the addition parameters that are added in stack attention?
>
> The SDPA sublayer function uses the parameters $\mathbf{W}\_{\mathrm{q}}, \mathbf{W}\_{\mathrm{k}} \in \mathbb{R}^{d\_k \times d\_{\mathrm{model}}}$, $\mathbf{W}\_{\mathrm{v}} \in \mathbb{R}^{d_v \times d\_{\mathrm{model}}}$, and $\mathbf{W}\_{\mathrm{y}} \in \mathbb{R}^{d\_{\mathrm{model}} \times d\_v}$. Stack attention replaces these with the parameters $\mathbf{W}\_{\mathrm{a}} \in \mathbb{R}^{d\_a \times d\_{\mathrm{model}}}$, $\mathbf{W}\_{\mathrm{v}} \in \mathbb{R}^{m \times d\_{\mathrm{model}}}$, and $\mathbf{W}\_{\mathrm{y}} \in \mathbb{R}^{d\_{\mathrm{model}} \times d\_r}$. Nondeterministic stack attention additionally adds $\mathbf{w}\_{\mathrm{v}} \in \mathbb{R}^m$, which is used for learning the initial bottom vector $\mathbf{v}\_0$. In the case of superposition stack attention, $d\_a = 3$ and $d\_r = m$. In the case of nondeterministic stack attention, $d\_a = |Q| \cdot |\Gamma| \cdot |Q| \cdot (2|\Gamma| + 1)$ and $d\_r = |Q| \cdot |\Gamma| \cdot m$.
>
> > The superposition stack works much better than the non-deterministic stack (both in terms of speed and results on natural language data) but as you said, it is unable to make multiple reduces at each token. Does this then mean that it cannot produce natural language syntax trees?
>
> You are correct that because of superposition stack attention's inability to perform multiple reductions per timestep, we expect that it would not be effective at modeling certain syntactic patterns that require closing multiple constituents at once, such as prepositional phrase attachment. However, it should still be advantageous for a subset of syntactic patterns where this is not an issue, such as center embedding (e.g. "The diamond that the thief who the man saw stole glittered") and gross syntactic expectation (e.g. an independent clause must come after a subordinate clause beginning with "As"). Note that although superposition stack attention does perform better than nondeterministic stack attention on our MT task, nondeterministic stack attention has better perplexity on language modeling. We think this is because of the difference in expressivity.

---

### Author Response · Authors · 2023-11-22
**Summary of Changes**

Dear reviewers,

Thank you very much for all of your valuable feedback. We have uploaded a revision of our paper that addresses many of your comments. Our changes can be summarized as follows:

* We have included an analysis of the stack actions learned by superposition stack attention on the $w\mathtt{\\\#}w^R$ and Dyck languages (Section 6 and Appendix E). Using heatmaps of the action probabilities, we show that the model has clearly learned the structure of the input. Since the set of stack actions in nondeterministic stack attention is much larger, analyzing nondeterministic stack attention and parses on natural language will require more advanced techniques, which may be pursued in future work.
* We list the parallel time complexity of each type of attention in the main text (Section 5), with a longer discussion provided in Appendix C.
* We provide more details about parallelizing nondeterministic stack attention in the style of parallel CKY, which would improve it from $O(n^2)$ to $O(n \log n)$ parallel time (Appendix C). The factor of $O(\log n)$ comes from a summation of size $O(n)$.
* We have simplified the exposition of the dVPDA (Section 3.2.2), moving certain details to the appendix (Appendix A). We now explain the size of $\Delta_t$.
* We have added a reference to Murty et al. (2023) (Section 1).
* We have made the language at the end of Section 1 more specific.

---

### Meta-Review · Area_Chair_GcuB · 2023-12-10

**Metareview:**

This paper presents a stack-augmented transformer to help overcome some of the challenges of ordinary self-attention in modeling nested syntactic structure.

This is an interesting idea and this paper presents a good way toward it. All the reviewers admit its contribution and it is a clear acceptance.

**Justification For Why Not Higher Score:**

n/a

**Justification For Why Not Lower Score:**

n/a

---

### Decision · Program_Chairs · 2024-01-16

Accept (spotlight)